# Development and Physico-Chemical Characterization of Healthy Puff Pastry Margarines Made from Olive-Pomace Oil

**DOI:** 10.3390/foods11244054

**Published:** 2022-12-15

**Authors:** María Dolores Álvarez, Susana Cofrades, Miriam Pérez-Mateos, Arancha Saiz, Beatriz Herranz

**Affiliations:** 1Institute of Food Science, Technology and Nutrition (ICTAN-CSIC), c/José Antonio Novais, 6, 28040 Madrid, Spain; 2Department of Food Technology, Veterinary Faculty, Complutense University, Avda/Puerta de Hierro, s/n, 28040 Madrid, Spain

**Keywords:** puff pastry margarine, olive-pomace oil, rheological properties, fatty acids profile, fat crystals

## Abstract

Due to its characteristic aroma and flavor, puff pastry (PP) prepared with butter is more accepted than that made with margarine, yet a high saturated fat consumption is associated with cardiovascular disease. This work studies the potential of olive-pomace oil (OPO) as main ingredient of PP margarines together with different organogelator agents to imitate the technological properties of a commercial fatty preparation (CFP). Rheological and textural properties were measured in all formulated margarines (FM) and, additionally, thermal and microstructural properties, and fatty acid (FA) profiles were analyzed in some selected FM. The different FM had viscous modulus (*G*″) and loss factor (tan *δ*) values lower than those of CFP, thus reflecting a different viscoelasticity and plasticity. The crystallization and melting temperatures of FM were also different from those of CFP, indicating the presence of a dissimilar polymorphic fat-crystal structure. Nevertheless, the FM containing an oleogel prepared with 5% beeswax and OPO was more similar to CFP. The FA profile of CFP and FM, with 80% polyunsaturated fatty acids (PUFA) and 60% oleic acid, is healthier than that of a PP commercial butter (CB), evidencing that, although improvements in margarine plasticity are still necessary, OPO is technologically viable to produce healthier PP margarines.

## 1. Introduction

Margarine and butter are well known fats (water-in-oil emulsions, W/O) often used to produce high-fat bakery products, such as puff pastry (PP). PP is an aerated and laminated product with a high fat content obtained by repeated lamination of wheat dough and fat [1]. Due to their characteristic aroma and flavor, and their high plasticity, PP doughs made with butter are more widely accepted than those made with margarine. The fat used should be solid and firm enough to separate the layers of dough during the lamination process. However, the high content of saturated fatty acids (SFA), as well as the high costs and difficult handling of butter during PP industrial processing, have promoted the development of commercial fatty preparations (CFP) and margarines from properly structured healthy vegetable oils and fats. These products present a better lipid profile and better processability for commercial PP production than butter [1]. To this end, technological strategies are providing new possibilities to develop healthier solid fats from vegetable oils (low content in SFA and high content in unsaturated fatty acids, UFA), more in line with current healthy intake recommendations [2], and with similar physical properties to butter.

Edible oil (rich in UFA and reduced in trans-fatty acids, TFA) structuring is mainly based on either oil gelation by the addition of suitable organogelator agents, such as triglycerides (TG), diglycerides (DG), monoglycerides (MG) and their mixtures, fatty acids (FA), fatty alcohols, waxes, etc., or on mixtures of high- and low-melting fats [3]. Most of the studies using organogelator agents to structure edible oils have focused on the production of chocolate paste, cakes, and spreadable or table margarines [3,4,5,6]. However, as PP margarines must present specific characteristics, such as high plasticity, a high content of total solid and enough firmness to withstand rolling and baking, the use of oil structuring to develop this type of margarine has been less studied, constituting a major technological challenge [7]. In addition, and due to its key role in PP, saturated fat cannot be entirely reduced without negatively affecting product quality [8]. For this reason, palm stearin (PS) is a useful hardstock and contributes to the formulation of margarines and shortenings by providing strength or structure to the products and improving plasticity [9]. Thus, in many of the new PP margarine formulations, PS constitutes a part of the fat/oil mixture. In this regard, it is important to note that both the composition (types and proportion of oils/fats and emulsifiers/organogelators) and the processing conditions of this type of margarine (mainly emulsification temperature and time, and cooling rate) determine the final quality of the margarines and, therefore, of the PP. Silow et al. [10] studied the impact of fat/oil blends with different amounts of TFA on the quality of conventional and reduced-fat PP, concluding that the blend with the highest proportion of TFA was the most suitable for PP products, presenting satisfying baking results. In addition, Lupi et al. [3,11] optimized the formulation of water-in-olive oil emulsions (olive oil and cocoa butter mixtures) structured with different proportions of an organogelator agent (Myverol 18–04 K, mainly composed of MG) to obtain PP.

With regard to the manufacturing process conditions, Calligaris et al. [7] proposed a gentle drying procedure at 30 °C to allow partial water removal from structured emulsions containing saturated MG, sunflower oil, and water. The authors concluded that this novel fat is a good candidate for both PP lamination and baking. Additionally Soronja-Simović et al. [12] investigated the influence of two margarines, different in fat-phase composition and physical properties, and determined the minimal amount of margarine necessary to form PP with a layered structure, as well as the optimal dough rest period. Different processing conditions to obtain oleogels (OG) from high-oleic sunflower oil and Myverol 18–50 XL PL were also explored to develop a fat with similar rheological and textural characteristics to those of margarine [13]. These authors observed that the rheological parameters were strongly dependent on the preparation variables, especially Myverol concentration and cooling temperature, and that the oleogels with the highest elasticity were more difficult to spread.

Olive-pomace oil (OPO), a by-product obtained after olive oil extraction, is an oil with health properties and potential technological applications in the production of PP margarine [14,15]. OPO has a lipid composition comparable to that of olive oil, but a different content of bioactive compounds, which affects its health-promoting potential [16].

Different studies suggest that OPO is a relevant source of monounsaturated fat in the diet, as well as an important dietary source of vitamin E, fulfilling the recommended daily intake [17]. Therefore, the regular consumption of OPO in the diet presents hypolipidemic effects in healthy consumers and in subjects with high blood cholesterol, contributing to the prevention of cardiovascular disease [18]. Also, in comparison with high-oleic sunflower oil and sunflower oil, the intake of OPO may improve cardiometabolic risk, as it reduces levels of cholesterol-related parameters and waist circumference in healthy and hypercholesterolemic subjects [19]. In addition, certain lipophilic bioactive compounds of OPO, in particular α-tocopherol and oleanolic acid, could have a protective effect against Alzheimer’s disease since they are transported into the brain via triglyceride-rich lipoproteins [20].

As regards processing quality, frying with OPO has also been shown to improve the nutritional properties of pre-fried potatoes, pre-fried battered chicken nuggets and chicken croquettes [21,22].

On the other hand, the food industry needs technologies and industrial processes that meet current demands in terms of nutritional quality and environmental care and that, at the same time, are economically profitable. In Spain, OPO is much more than a single food product; during its production, the pomace sector carries out great environmental work and it is an example of circular economy and zero waste.

For all the above reasons, the objective of this research was to study, for the first time, the potential applicability of OPO as the main ingredient of margarine for the production of healthy PP doughs. For this purpose, different suitable organogelator agents (beeswax, Palsgaard 6111^®^ and Verol N-90) were chosen and mixed with OPO in the oil phase (OP) to obtain formulated margarines (FM) with structural features close to those of a commercial fatty preparation (CFP). In addition, aqueous phase (AP) gelators agents (gelatine and transglutaminase) were also tested. Rheological and textural properties were measured in all the FM, analyzing thermal properties, microstructure, and the FA profile as well in some selected FM.

## 2. Materials and Methods

### 2.1. Materials

OPO was supplied by INTERPROFESIONAL del Aceite de Orujo de Oliva (ORIVA, Sevilla, Spain). Refined palm stearin (PS) in flakes (79% of SFA) was kindly donated by Vandemoortele Europe NV (Ghent, Belgium). Organogelators Palsgaard^®^ 1311 and Palsgaard^®^ 6111 were generously provided by Palsgaard A/S (Juelsminde, Denmark). Palsgaard^®^ 1311 is a mixture of citric acid esters of MG and DG of vegetable FA and polyglycerol esters of FA, used as an emulsifier for PP margarine. In turn, Palsgaard^®^ 6111 is composed of TG from fully hydrogenated canola oil, commonly used in margarine to favor the crystallization process. Verol N-90, distilled MG (E-471), Verol P, polyglycerol ester of FA (E-475), Verolec Non GMO IP, made of soybean lecithin (E-322), and beeswax were all kindly donated by Lasenor (Barcelona, Spain). Ground dietary gelatin 200/220 Bloom was acquired from Manuel Riesgo S.A. (Madrid, Spain). A preparation based on transglutaminase (MTG) (ACTIVA EB^®^) was kindly donated by Ajinomoto Foods Europe (Paris, France). Salted cow’s milk butter from Cooperativa Ganadera del Valle de los Pedroches (COVAP, Córdoba, Spain) was purchased at a local supermarket. A comercial fatty preparation (CFP) and a commercial butter (CB), both for PP, and used as controls in this study, were acquired from St. Auvent^®^, Vandermoortele Europe, Ghent, Belgium) and Flechard SAS Laiterie Du Pont Morin (France), respectively. Butter flavor was bought in Arconsa S.A. (Murcia, Spain). Citric acid anhydrous (Panreac, Barcelona, Spain) and salt (fine table salt) (Aliada, Madrid, Spain) were also used. All different ingredients and fatty additives were kept under adequate conditions to protect them from oxidative stress and used without further purification.

### 2.2. Margarine Preparation

The FM were prepared in batches of 600 g. The composition of the nine FM, coded as FM1-FM9, is shown in Table 1. In all of them the oil phase/aqueous phase (OP/AP) proportion was close to 80/20. To prepare the AP, all ingredients were mixed and stirred using a magnetic stirrer at 300 rpm for 10 min at room temperature until completely dissolved. In all the FM, the ingredients of the AP were the same, in accordance with the AP composition of CFP, with the exception of FM3 and FM4, which, in addition, contained 1.5% gelatin or 0.75% MTG, respectively.

The choice of the organogelators that constituted the OP was based on the different combinations and percentages recommended by the PP margarine manufacturers (Palsgaard^®^ and Lasenor) with slight modifications. In all the FM, the percentages of PS (27.5%), Verolec-Non GMO IP (1.5%), Palsgaard 1311^®^ (1.2%), and Verol P (1.5%) were fixed. However, the incorporation of Palsgaard 6111^®^ at different concentrations (0.70–0.90%) or 0.83% Verol N-90 was studied due to their suitable plasticizing and stabilizing properties in PP margarines. To prepare the OP, its components were mixed at 300 rpm with a rod stirrer (Bunsen AGV-8, Madrid, Spain) by placing them in a thermostatic water bath (Series BD, Bunsen, Madrid, Spain) at 70 °C. Once the OP was dissolved, the temperature was reduced to 50 °C, by introducing the mixture into a Thermomix^®^ glass. As the main purpose of this work was to study the effect of composition, the conditions of the emulsification process and subsequent cooling were kept constant. The emulsification was performed at 45–50 °C using a Thermomix^®^ TM5-1 homogenizer (Vorwerk, Germany) by incorporating the AP into the OP, at 1100 rpm, and then increasing the stirring speed to 4400 rpm for at least 13 min. Then, the emulsion was rapidly cooled under gentle stirring by placing it in a cold ice bath (5 °C) for 4 min until a final temperature of 25 °C was reached. Finally, each emulsion was poured into a plastic tray and left to rest for 48 h at room temperature before use. FM5 and FM6 (Table 1) included in their composition an OG prepared with 5 and 3% beeswax and 35 and 37% OPO, respectively. Both OG were elaborated at 80 °C for 24 h before their incorporation in the OP.

### 2.3. Rheological Measurement

Rheological characterization of the controls (CFP and CB) and the FM was carried out with a Kinexus pro controlled-stress rheometer (Malvern Instruments Ltd., Worcestershire, UK), equipped with a serrated plate–plate geometry (Ø = 20 mm, 1.5-mm gap, ≈30 mL sample). The temperature was controlled with a high-temperature cartridge in the lower plate, and a cover cell to maintain the samples at a specific temperature and prevent evaporation. Before further testing, each sample was allowed to stabilize and restructure for 20 min at 20 °C using a time sweep test performed at 1 Hz with a selected shear stress (*σ*) within the linear viscoelastic region (LVR) (*σ* = 20 Pa for CB, FM2 and FM4, and *σ* = 200 Pa for CFP and the rest of FM). Following this, stress sweep tests were performed at 1 Hz and at 20 °C with *σ* varying from 2 to 2000 Pa, depending on sample consistency, in order to determine the limit values of the LVR. Then, frequency sweep tests were carried out at 20 °C by the harmonic variation of strain with time at a frequency range 0.1–10 Hz, and a selected stress amplitude within the LVR (mostly *σ* = 200 Pa). The weak-gel model: |*G**| = [(*G*′(*f*))^2^ + (*G*″(*f*))^2^]^0.5^ = *Af*^1/z^, where *f* is the frequency, *z* the network extension, *A* is the strength of the interactions, and *R*^2^ is the determination coefficient, was used to interpret the mechanical spectra [3]. In addition, temperature sweep tests were carried out from 10 to 50 °C, at a heating rate of 1 °C/min and at 1 Hz. Again, the stress amplitude was mostly set at *σ* = 200 Pa, in order to stay within the LVR at the beginning of the heating process. From all the different tests performed, elastic modulus (*G*′, kPa), viscous modulus (*G*″, kPa), complex modulus (*G**, kPa), phase angle (*δ* °), and loss tangent (tan *δ* = *G*″*/G*′, dimensionless) values were recorded.

### 2.4. Texture Measurement

A TA.HDPlus Texture Analyzer (Stable Micro Systems, Ltd., Godalming, UK) provided with Texture Exponent software (version 6.1.20.0) and equipped with a 5 kg load cell was used. The texture of CFP, CB, and FM was instrumentally measured at 20 °C by means of two penetration tests (flat and conical) carried out with a 4-mm cylindrical flat stainless-steel probe (P/4; Stable Micro Systems Ltd., Godalming, UK) and with a 30-mm diameter conical methacrylate probe, respectively. Both probes penetrated the sample to 10 mm at a rate of 1 mm/s. From force–distance curves of flat penetration, force (N) at 10 mm, maximum force (N), and penetration work (mJ) were derived. In turn, from force–distance curves of conical penetration, force (N) at 10 mm, and penetration work (mJ) were calculated.

### 2.5. Differential Scanning Calorimetry

Melting and crystallization profiles of PS, CFP, CB, and three different FM (FM5, FM7 and FM9) were measured using a differential scanning calorimeter (DSC) (TA Q1000, TA Instruments, New Castle, DE, USA). An empty, hermetically-sealed aluminum pan was used as reference, and 6–8 mg of the sample was placed into an aluminum pan with lid and hermetically sealed. For cooling thermograms (exothermic response), the sample was equilibrated at 80 °C for 10 min, and then cooled from 80 to −50 °C at a constant rate of 5 °C/min. In turn, for heating thermograms (endothermic response), the sample was kept at −50 °C for 10 min, and then heated from −50 to 80 °C at 5 °C/min. Cooling and heating thermograms were recorded and crystallization and melting peak temperatures (*T*_cp_ and *T*_mp_) were calculated by means of the Data Analysis 2000 software (v. 4.1D, TA Instruments, New Castle, DE, USA).

### 2.6. Fatty Acid Profile

Fatty acid (FA) contents were determined by saponification and bimethylation as described by Alvarez et al. [23]. All the samples (PS, CFP, CB, FM5 and FM8) were previously lyophilized. Fatty acid methyl ester (FAME) was analyzed on an Agilent gas chromatograph (Model 7820A, Santa Clara, CA, USA) fitted with a GC-28 Agilent DB-23 capillary column, and a flame ionization detector was used. FA were identified by comparing retention times with an FA standard (Supelco 37 FAME Mix 47885-U, Bellefonte, PA, USA). An internal standard C13:0 was used for quantification and added to the sample before methylation, and the results were expressed as mg FA/g sample.

### 2.7. Polarized Light Microscopy

The microstructure of CFP, CB, and four FM (FM2, FM4, FM5 and FM8) were visualized by polarized light microscopy on a Leica AF6000 LX system microscope (Mannheim, Germany) in a Pecon environmental chamber kept at 37 °C (Promi III Pol, Carl Zeiss AG, Germany) and images were captured using a digital camera Hamamatsu C9100-02. The samples were heated at 50 °C for 1 h. Then, 15 mg of each sample were placed in the center of a glass slide and covered with a coverslip. The slides were allowed to cool and crystallize at 5 °C for 10 days and then the microstructure was analyzed. Images were captured using a 10×/0.30 NA and a 40×/0.75 objective lens with 1.6× magnification added and with the Hg-arc lamp. Images were stored with a resolution of 1000 × 1000 pixels using the Leica Application Suite X (LAS X) software (Leica, Johnson City, TN, USA).

### 2.8. Statistical Analysis

All measurements mentioned above were repeated at least in triplicate and performed 48 h after the formulation of the margarines, with the exception of polarized light microscopy. One-way analysis of variance (ANOVA) was used to compare rheological, textural, and thermal parameters of CFP, CB, and the FM. Significant differences between pairs of means were evaluated by the Tukey test, using a 95% confidence interval (*p* < 0.05). Analyses were performed using IBM SPSS for Windows, Version 27.0 (IBM Corp., Armonk, NY, USA).

## 3. Results and Discussion

### 3.1. Rheological Measurement

#### 3.1.1. Time Sweep Test

As an example, Appendix A shows time sweeps carried out in the LVR corresponding to samples CFP, CB, FM8, and FM9 to allow restructuring and stabilization after their placement in the rheometer. *G*″ values of FM9 overlap with *G*′ values of CB. During this test, *G*′ values always increased, whereas *G*″ usually decreased. Furthermore, the percentage variation of *G*′ was generally much higher than that of *G*″. For example, in CFP, *G*′ increased by 12% and *G*″ decreased by 3%, whereas in the FM the increase of *G*′ ranged between 7 and 22% and the decrease of *G*″ varied between 0 and 12%, depending on the FM measured. However, in CB both *G*′ and *G*″ increased by 24 and 11%, respectively. This evolution would be associated with reorganization, recovery, and even with a possible structure crystallization of diverse crystalline entities, that is, of mixtures of lipids, mainly palmitic acid (C16:0) and oleic acid (C18:1n9), which are present in the samples according to their different composition, and that would have suffered a destabilization process. 

Therefore, given the nature of the systems studied and the rapid evolution of their viscoelastic properties over time, time sweeps reflect the importance of allowing a complete restructuring of each sample before proceeding to the subsequent characterization of its rheological behavior.

#### 3.1.2. Stress Sweep Test

Stress sweeps to determine the LVR of CFP, CB, and different FM (FM1, FM2, FM3, FM4, FM5 and FM7) are shown in Figure 1. Note that stress ranged from 2 up to 200 Pa and from 20 up to 2000 Pa in softer and harder samples, respectively. *G*′ values of CFP overlap with those of FM3 and FM7, although the control CFP had higher *G*″ values. In turn, at 20 °C, CB exhibited the lowest *G*′ and *G*″ values, and the highest phase angle (*δ*) values. 

Table 2 shows the limit or critical values of the LVR for all samples. FM5 and FM7 had the highest critical shear stress (*σ*_max_) values, which reflect that the structures of these formulations containing 35 and 40% OPO, respectively (Table 1), were tighter, being constituted by a denser crystalline network. In the case of FM5, this result could be attributed to the presence of an OG (prepared with OPO and beeswax at 5%) and MG from Verol N-90 (Table 1) among its ingredients. Previous studies have shown that olive oil and a virgin olive oil-beeswax OG may be used as plastic fat replacers in breakfast margarines [21,24,25]. In fact, Silva et al. [26] reported that the use of “waxy” organogels in margarines produced a strong network with an excellent oil retention capacity, generating margarines with hardness and rheological parameters similar to those of commercial margarines. These authors used candelilla wax and MG and demonstrated a synergistic effect of both ingredients in the structuring and emulsification of the margarines due to the incorporation of MG in the crystalline network, which improved the network organization [27]. In turn, FM7 did not contain MG, but it contained 0.80% Palsgaard 6111^®^ that consists of fully hydrogenated TG and prevents the oiling out phenomenon, thus it is often used as an oil absorber and crystal starter in chocolate spreads and fat products [28]. Although *σ*_max_ values of FM6 and FM8 (which contained OPO-beeswax at 3% OG and 0.90% Palsgaard 6111^®^, respectively) were close to that of CFP, the value corresponding to the control CFP was significantly lower than those of FM5, FM6, FM7, and FM8 (Table 2). Moreover, these four FM showed the highest values of fat-crystal rigidity (*G**_max_), being FM7 the margarine that had the most similar value to that of CFP. In contrast, FM9 had the lowest *σ*_max_, which could be associated with its lower TG content (Table 1).

Regarding the conformational flexibility (*γ*_max_) of the different samples, there were no significant differences between the value corresponding to CFP and those of the margarines containing Verol N-90: FM3, FM4, FM5, FM6, and FM8. Verol N-90 is a distilled MG made from fully hydrogenated palm oil. It is widely used as an emulsifier, hence, it reduces interfacial tension between the water phase and the fat phase in emulsions, providing a fine water droplet dispersion. Lupi et al. [3] reported that oil crystallization is mainly related to MG presence, which contributes to the creation of a well-organized fat-crystal network.

In addition, the highest values of *γ*_max_ corresponded to FM7, FM9, and CB, these being the most flexible samples. Moreover, FM7 showed network rigidity (*G**_max_) equal to that of CFP. On the other hand, CB together with FM2 and FM4 were the softest samples, presenting the lowest *σ*_max_ and *G**_max_ values (Table 2), which could be related to the fact that both FM2 and FM4 did not contain Palsgaard 6111^®^ nor beeswax. However, it seems that when these two ingredients are added together (FM1), they do not present a synergistic effect; the values of *σ*_max_ and *G**_max_ are significantly lower than those corresponding to CFP, FM5, FM6, FM7, FM8, and FM9, yet higher than those of CB, FM2, and FM4. On the other hand, FM3 had the same OP formulation as FM2 but with less water, and with 1.5% gelatin in the AP, which provided stiffness to the sample (Figure 1).

Finally, CFP and CB presented a tan *δ*_max_ significantly higher than that of the FM, thus evidencing the presence of a network structure with a lower viscoelasticity or a lower degree of solidity.

#### 3.1.3. Frequency Sweep Test

As an example, the mechanical spectra obtained for the different FM in comparison with those of CFP and CB are shown in Appendix A. Apparently, all FM show a rather similar rheological behavior, *G*′ values being higher to those of *G*″ in the *f* range analyzed. Additionally, both moduli, in particular *G*′, showed applied frequency dependence which, according to the definition of a gel, indicates that all samples present a rheological behavior of “structured liquid” (formerly called of weak gel) [29,30]. In CFP, *G*′ and *G*″ values increased by 44 and 20%, respectively, with the increase of *f*. Likewise, CB presented a behavior with respect to *f* quite similar to that of CFP (*G*′ and *G*″ values increasing by 50% and 16%, respectively). However, in most FM, *G*″ values decreased with increasing applied frequency. For instance, in FM2, *G*′ and *G*″ increased and decreased by 39 and 10%, respectively, with the increase of frequency. This was also the case for FM9, where *G*′ and *G*″ values increased and decreased with increasing frequency, by 18 and 6%, respectively. However, in FM1, and as observed in CFP, the values of both moduli increased with increasing frequency, although in this sample it was only by 32 and 18%, respectively. 

Additionally, Table 3 presents *G*′, *G*″, and tan *δ* values corresponding to 1 Hz frequency for all samples. FM8 exhibited the highest *G*′, although *G*″ and tan *δ*_max_ values in this sample were significantly lower than those of CFP. However, CFP had the highest *G*″, whereas CFP and CB, FM1, FM2, and FM4 presented higher tan *δ* values (>0.1), which would reflect the lower degree of solidity of all of them. The rest of the FM presented tan *δ* < 0.1, indicating a more elastic character. 

All FM can be considered structured emulsions with a weak gel behavior (currently known as structured liquids) [29], as a result of the development of a three-dimensional network of molecules and fat crystals that form weakly connected rheological “units” [31]. In these cases, the structured OP traps the oil and water droplets, producing a stable W/O emulsion. The linear trend observed in the log–log mechanical spectra for both moduli (Appendix A), allows the rheological data to be interpreted using the “weak gel” model proposed by Gabriete et al. [31]. The results of these adjustments are also shown in Table 3. A high *A* value would indicate an increase in interaction forces within the network, while a high *z* value suggests a large number of interacting units, thus increasing the connectivity of the network [3,31]. At 20 °C, FM8 exhibited significantly higher A and *z* values than the rest of the FM and the control CFP, which would imply a larger number of interactions within the network and a much higher degree of crystallization. These greater values of A and z could be related to a higher *G**_max_ and a low *γ*_max_ (Table 2), suggesting that this sample would have a very rigid network with a significant number of structural units (crystalline regions) strongly connected to each other. These tightly packed crystalline regions would increase the strength of the lattice, resulting in a rigid and brittle structure [32]. Additionally, as it is observed in Table 3, the softer margarines, FM1, FM2, and FM4, with a lower degree of solidity (tan δ > 0.1), presented a lower number of interacting structural units (lower *z* values), closer to those of the controls CFP and CB. However, these units do not appear to be sufficiently interconnected with each other (low *A* values) to provide a network with a structure similar to that of CFP. Regarding the *z* value obtained in CFP, it was much lower than the corresponding value in the rest of the FM. Other authors [11] have reported an *A* value of 43,000 ± 200 Pa in commercial margarine, which is much lower than that of the different FM in this study, and even lower than the corresponding one in CB (Table 3). Yet, the *z* value of that margarine (14.0 ± 0.6) was quite similar to that obtained in CFP (12.8 ± 0.24). Moreover, this lower *z* value, together with that of CB, is probably a consequence of the fact that G’ and *G** are more dependent on the applied frequency in both controls than in FM, which means there is a lower connectivity between the crystalline entities or polymorphs present in the three-dimensional network of both controls.

#### 3.1.4. Temperature Sweep Test

Examples of temperature sweeps for CFP, CB, and different FM carried out between 10 and 50 °C are shown in Figure 2. In turn, Table 4 shows the crossover temperatures of the viscoelastic moduli obtained in all the different samples. *G*′ values were slightly higher at 10 °C, in response to the more complete crystallization that fat crystals exhibit at lower temperatures, and the viscoelastic moduli gradually decreased with increasing temperature until the crossover of *G*′ and *G*″ occurred, between approximately 35 and 41 °C. However, in CB, the crossover temperature was much lower (30 °C). This result would reflect that butter presents a crystalline network that, although it seems to have a lower degree of connectivity than that of the rest of the margarines (lower A and *z* values), would actually be formed by polymorphs comprised of different FA which, consequently, display a different organization [33]. Furthermore, FM2 and FM4, which had a more weakly connected network, presented a lower crossover temperature than the rest of the FM (Table 4).

Nevertheless, in all samples, the crossover of both moduli occurs as a consequence of the melting of the fat crystals present in the three-dimensional network; at the moment of the crossover, all samples no longer presented a solid-like rheological behavior, but instead, they exhibited a behavior similar to that of a liquid (*G*″ > *G*′). The crossover between *G*′ and *G*″ occurred at 41.0 °C in CFP (Table 4), and both FM6 and FM7 presented crossover temperatures significantly equal to those of CFP. FM6 and FM7 were formulated with the same AP (Table 1), although FM6 contained less OPO in its OP (37%) than FM7 (40%). As well, FM6 included an OG with beeswax (3.0%) and Verol N90 (0.83%) instead of the emulsifier Palsgaard 6111^®^ (0.80%) that was present in FM7. Similar results were obtained by Lupi et al. [3] in W/O emulsions studied as healthy shortening replacers. These authors performed temperature sweeps from 70 to 20 °C, using two different cooling rates (1 and 5 °C/min), and also found critical temperatures for fat crystallization onset above 42 °C. On the other hand, and according to Fomuso and Akoh [34], the temperature corresponding to the structural melting of the fat obtained by dynamic testing, can be used to predict the butter melting perception that the consumer will experience.

### 3.2. Textural Measurement

Figure 3 shows force–distance curves obtained from flat (Figure 3a) and conical penetration tests (Appendix A) performed on two FM (FM5 and FM8) in comparison with those obtained when testing CFP and CB. FM containing OPO should exhibit force–distance profiles as similar as possible to those obtained for CFP, which would reflect that their structural organization is also similar. 

Firstly, it may be observed that the curves generated for each type of penetration test are very different. When penetrating with the flat tip (Figure 3a), CFP shows a force–distance profile with a smooth initial fracture peak, and no other force peaks are seen when the probe deforms the sample, indicating that CFP has a structure with very high plasticity [35]. Furthermore, the maximum penetration force is reached at approximately 5 mm, and does not coincide with the force at 10 mm. Likewise, CB and FM5 presented pronounced plasticity, as it is shown by their corresponding curves in which no fracture peak is detected, with the force increasing as the probe penetrates, reaching the maximum value at 10 mm (Figure 3a). However, in FM8, it is possible to identify an initial force peak, which was reached almost instantaneously at the beginning of the test, concurring with the maximum force, (Figure 3a), as well as some other force peaks in the profile. This would indicate that FM8 has high fracturability, as shown by the jagged shape of its force–distance curve, suggesting that this deformation also induces the breakage of some β crystals [36]. Thus, this reveals that FM8 presents a rigid and brittle lattice with very little flexibility, probably due to the tight packing of crystalline units found strongly connected by weak interactions, as discussed above. 

In turn, when performing the conical penetration test (Figure 3b) small force peaks were observed in FM8, indicating the breakage of the crystalline entities, also reflecting its brittle character. In contrast, in CFP, FM5, and CB, the force–distance profiles of the conical penetration test presented no force peaks, and increased continuously with the distance travelled by the probe, reaching the maximum force at the end of the test.

Table 5 shows the texture parameters derived from both tests in the different FM. The force values measured at 10 mm and corresponding to FM5 and FM6, both formulated with beeswax as the structuring agent, were significantly higher than that obtained for CFP, reflecting that the hardness or consistency of these margarines may be excessive for proper handling and lubrication of the ingredient mixture in the PP doughs. The margarine with values of force at 10 mm and maximum force closest to those of CFP was FM8, however, these were significantly lower and higher, respectively, probably due to its brittle nature. The highest penetration work was also obtained in FM5 and FM6, and although there were significant differences, these values are quite similar to those of CFP. 

Additionally, the maximum force derived from the conical penetration test or spreadability test, which in this case coincided with the force at 10 mm (Figure 3b), can be considered a measurement of sample firmness, whereas the area under the curve corresponding to the total force required is referred to as spreadability, in spreads such as margarines [26] and pâtés [37]. At 20 °C, the sample presenting the highest firmness was FM5, followed by CFP, with firmness values significantly equal to those of FM6 and FM8 (Table 5), while the spreadability of CFP was also similar to that of FM3. As previously mentioned, this may be due to the fact that FM8 exhibited some firmness peaks (Figure 3b) and would therefore be too brittle, while, on the contrary, FM3 exhibited no force peaks during its penetration, indicating a more adequate plasticity that would also be similar to that of CFP. This could be related to the presence of gelatin in its AP and MG in its OP, and to the absence of Palsgaard 6111^®^ in its formulation (Table 1). On the other hand, FM5 showed very high penetration work, and is quite far from that of CFP (Table 5). A higher spreadability value indicates a poorer spread of the margarine, and this hinders its proper incorporation into the initial dough in PP production. Thus, FM5 showed the highest values of texture parameters, which could be linked to the high values of critical rheological properties (*G**_max_ and *σ*_max_) observed on the LVR in this margarine (Table 2). Again, this result would indicate that the oleogel formed by 5% beeswax and OPO resulted in a strong and well-organized crystalline network. According to Lupi et al. [3], the hardness/rigidity of margarines is associated with a high melting temperature of the saturated triglycerides present in their OP, as they arrange in crystal aggregates forming an ordered interacting structure that surrounds and stabilizes water droplets. 

### 3.3. Thermal Properties

Crystallization and melting behaviors were investigated in CFP, CB, FM5, FM7, FM9, and in PS, as it was the second main ingredient of the OP (Table 1). Appendix A shows the cooling crystallization profiles of PS, CFP, and FM7 (Appendix A), as well as the complete CFP thermogram, thus including the melting profile (Appendix A). PS and CFP crystallization profiles were quite similar (Appendix A) and exhibited two clearly differentiated peaks. In both samples, a first peak is detected at a low temperature, around 0 °C, and a second major peak is located above 30 °C in the case of PS, and between 20 and 30 °C in CFP. In addition, although FM7 did not exhibit the first crystallization peak as clearly, it showed a very distinct peak between 20 and 30 °C, resembling CFP. PS was formulated with vegetable oils and fats (palm), which are responsible for the thermal behavior observed. More specifically, the second highest peak would be attributable to the solid fraction, i.e., palm fat and saturated TG [38], and the similarity of the thermal profile of CFP with that of PS would be ascribed to its high content of vegetable oils and fats, including palm fat. As well, the exothermic peak detected in FM7 profile, lower than that observed in PS and higher than that of CFP, would also be attributable to the presence of 27.5% PS in its formulation (Table 1).

CFP also presented two endothermic peaks (Appendix A); the first was found between 0 and 10 °C, and the second between 40 and 50 °C, separated by an intermediate exothermic peak between 15 and 20 °C, as described in previous studies [38,39]. Additionally, it was also possible to appreciate a third smaller peak around 50 °C, which would be explained by the presence of palm fat in its composition. In turn, the first endothermic peak detected in CFP would be associated with the liquid fraction (olein) and unsaturated triglycerides, while the second peak (between 40 and 50 °C), would be ascribed to the solid fraction and saturated triglycerides [38,39]. Moreover, the presence of a fine peak around 5 °C suggests the existence of β’ crystals in the CFP network [36], which are desirable as they provide margarine with a homogeneous consistency, and suitable gloss, color, and creaminess. The TG composition of the fat explains the diversity of polymorphic structures present in the margarines, the main and most stable crystalline forms being α, β and β′ [40]. Nonetheless, the OP of the margarines should preferably crystallize in the β′ form, as this confers a smooth mouthfeel, provides hardness to the final margarine and, in addition, traps a large amount of liquid oil due to its spherulitic nature [41]. Likewise, Garti et al. [42] mention that, when in presence of food emulsifiers, margarines are stabilized by microcrystals in the α form mixed with the β′ form.

The peak temperatures derived from the crystallization and melting thermograms are shown in Table 6. In both PS and CB, two crystallization peak temperatures (*T*_cp_), and three peak melting temperatures (*T*_mp_) were detected, although these temperatures considerably differed from each other. When compared to the other samples, the highest *T*_cp1_ and lower *T*_cp2_ in CB are in fact noticeable and associated with its high content of saturated dairy fat. In turn, and compared with CFP, in FM5, FM7, and FM9, the first *T*_cp1_ peak was not detected. Furthermore, the *T*_mp1_ of FM5, FM7 and FM9 were negative, while that of CFP was positive. As CB also had a negative *T*_mp1_, this result could be ascribed to the presence of butter (7.5%) in formulations FM5, FM7, and FM9 (Table 1). However, this first negative melting temperature has also been associated with the presence of MG in the formulation of margarines [38]. 

On the other hand, the *T*_cp2_ of the different margarines were statistically compared, and although the *T*_cp2_ of FM7 was significantly higher than that of CFP, no significant differences were found between the *T*_cp2_ of FM5, FM9, and CFP. As for the *T*_mp2_ of CFP, it was found to be significantly higher than that of the three FM, however, the differences are quite small. Therefore, the similar results obtained by DSC would evidence that the formulated margarines present, at least, some of the fat-crystal polymorphic structures present in CFP.

Finally, it should be noted that temperatures obtained at the crossover of *G*′ and *G*″ by temperature scans (Table 4) were only slightly lower than the *T*_mp2_ obtained by DSC and followed the same trend among them. This result confirms that temperature sweeps can definitely be considered adequate to study the melting of fat crystals present in margarines and other spreads.

### 3.4. Fatty Acid (FA) Profile

The FA profiles of PS, CFP, CB, and the selected margarines (FM5 and FM8) are shown in Table 7. As the lipid composition of PS and CB is markedly different, the FA of the five samples were not compared by statistical analysis. While in PS and CB, SFA are the most prevalent, in CFP, FM5, and FM8, the majority of FA are monounsaturated. The high content of SFA in PS (70%) is justified by its very high content of palmitic acid (61%), while the amount of stearic acid is also rather high (7%). However, in CB, the amount of C16:0 present is much lower than in PS (28%), quite similar to that of FM5 and FM8 (29%), and somewhat higher than that of CFP (22%). Therefore, it should be noted that although the formulated margarines contain PS in a high percentage (27.5%; Table 1), like table butter (7.5%), their palmitic acid content is only slightly higher than in CFP. In CB, in addition to C16:0 and C18:0, other short-chain FA along with myristic acid (9.5%) explain its high SFA content.

On the other hand, the FA profile of CFP is substantially healthier than that of PS and CB, its content in oleic acid (C18:1n9) being more than double (39%), and very similar to that found in FM5 and FM8 (40 and 41%, respectively). In FM5 and FM8, the abundance of C18:1n9 is clearly attributable to the OPO present in the formulations. Furthermore, and regarding the total amount of monounsaturated fatty acids (MUFA), the differences between the contents for CFP, FM5, and FM8 are quite small, and all three samples have more than twice as much of the amount of MUFA in their lipid profile as PS and CB. As for PUFA, while the content of linoleic acid is higher than 7% in CFP, FM5, and FM8, in PS and CB, it is only found in percentages of approximately 4 and 1%, respectively. Consequently, the percentage of unsaturated fatty acids (UFA) is close to 50% in CFP, FM5, and FM8, and much higher than that quantified in CB (20%).

These results evidence that PP containing either CFP or one of the FM (FM5 or FM8) among its ingredients, will present a much healthier lipid profile for the consumer than the same PP elaborated with CB instead. 

### 3.5. Microstructure (Polarized Light Microscopy)

Appendix A contains polarized light microscopy (PLM) images of the crystalline structure of CFP (Appendix A), CB (Appendix A), and margarines FM2, FM4, FM5, and FM8 (Appendix A). Although it is not possible to identify the type of polymorphs or crystal forms present, the crystal lattice of all samples is formed by a homogeneous and dense mixture of small platelet-shaped crystals, moistened by a large amount of liquid oil that, according to other authors, could correspond to the β′ form [36,43]. Additionally, neither granular crystals nor larger spherical aggregates were observed in any of the margarines. It is worth mentioning that the presence of these granular crystals negatively affects the quality of margarine, as this granularity might be detected by the consumer [43]. However, this crystal growth occurs mainly during the conservation and/or maturation of margarine at 5 °C. In Appendix A no important differences can be observed in relation to the microstructure of samples CFP, FM2, FM4, FM5, and FM8. Yet, according to the rheological results, the connectivity of the interactions in the three-dimensional network corresponding to CFP should be lower (a significantly lower value for parameter z, Table 3) than that of the rest of the margarines. Conversely, it is possible to appreciate that CB exhibits a finer and less dense fat-crystal network (Appendix A) and is made up of smaller fat crystals than those observed in the margarines. This result is attributed to the milk-based lipid composition of butter. In any case, although different primary TG crystalline forms (α, β y β′) may be present in the structure of the margarines, it would be necessary to conduct different analyses, such as X-ray diffraction, to discern the type of predominating polymorphs, in the margarines and in CFP [36].

## 4. Conclusions

The results of this study show that it is possible to formulate healthy PP margarines with OPO as their main ingredient (35.4–40.0%) using different combinations of organogelator agents. These FM, as well as CFP, exhibited a better lipid profile than CB. CFP and the FM presented 51 and 33% less SFA, respectively, and 60% more UFA than CB. Rheological measurements confirmed that all the FM behaved as a “structured liquid”, presenting lower, similar, and also higher elasticity (at 1 Hz) than CFP, as it was relatively feasible to reproduce the elasticity of CFP at a constant frequency by modifying the composition. However, it is more complex to formulate a margarine with elasticity that exhibits a frequency dependence similar to that of CFP, thus further studies should be conducted in future formulations to increase this dependency, which, in turn, will also have a positive impact on plasticity. Similarly, the fat-crystal aggregate structure of all the FM showed lower viscosity and higher viscoelasticity than CFP. The viscoelasticity of margarine seems to be a rheological property that determines its plasticity, and it is very difficult to imitate. Therefore, in order to improve the plasticity of margarine and reproduce more reliably the melting temperature of CFP, future formulations should be aimed at increasing their viscosity (>>*G*″), thus reducing their viscoelasticity (>>tan *δ*). This increase in plasticity also implies that the interactions in the fat crystal network should be weaker (<<*z* values), and in this regard, emulsion crystallization seems to be a determining factor, as it mainly affects viscosity. The crystallization and melting temperatures obtained by DSC reflect that the FM present some polymorphic structures of fat crystals similar to those of CFP. However, these FM lack a first crystallization peak, and exhibit a negative first melting temperature, therefore, it is also necessary that in future formulations the first melting approaches 5 °C, which would indicate the formation of more *β’* crystals, associated with higher plasticity.

Nonetheless, among the different FM, FM5, which contains an OG (prepared with 5.0% beeswax and OPO) and 0.83% MG from Verol N-90 was found to be the most suitable PP margarine, and quite similar to CFP. It displayed high and adequate plasticity, probably due to the presence of a well-organized fat-crystal network, but it presented higher firmness and lower spreadability than CFP. Therefore, this study demonstrates the technological applicability of OPO to formulate PP margarines. However, some improvements in the formulation and/or processing conditions are still necessary in order to reproduce and imitate more exactly all the rheological, textural, and thermal properties of the CFP polymorphic structure and, consequently, the technological properties required for its use in PP dough. 

## Figures and Tables

**Figure 1 foods-11-04054-f001:**
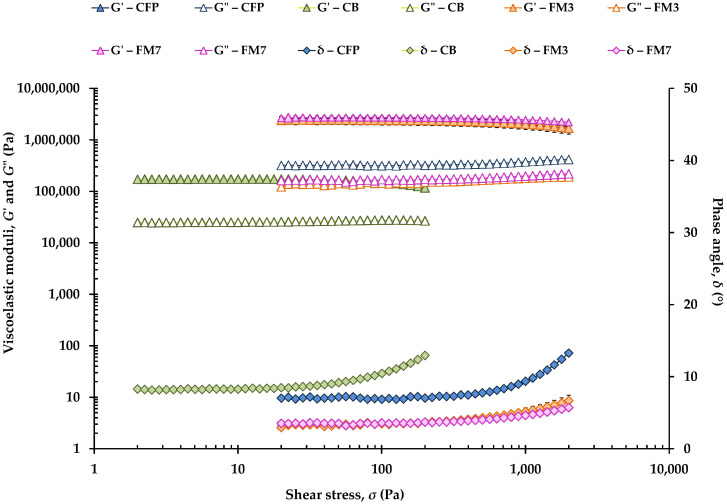
Stress sweeps carried out at 1 Hz and at 20 °C in a commercial fatty preparation (CFP), a commercial butter (CB) and two formulated margarines (FM3 and FM7). *G*′, elastic modulus; *G*″, viscous modulus.

**Figure 2 foods-11-04054-f002:**
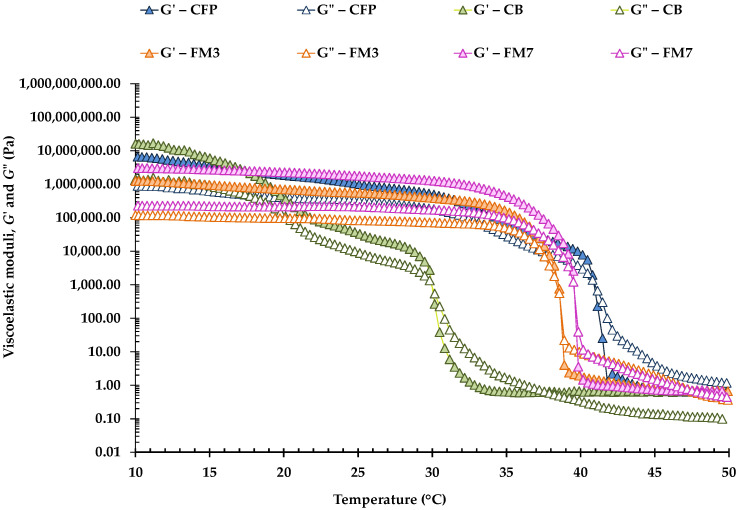
Temperature sweeps carried out at 1 Hz in a commercial fatty preparation (CFP), a commercial butter (CB) and two formulated margarines (FM3 and FM7). *G*′, elastic modulus; *G*″, viscous modulus.

**Figure 3 foods-11-04054-f003:**
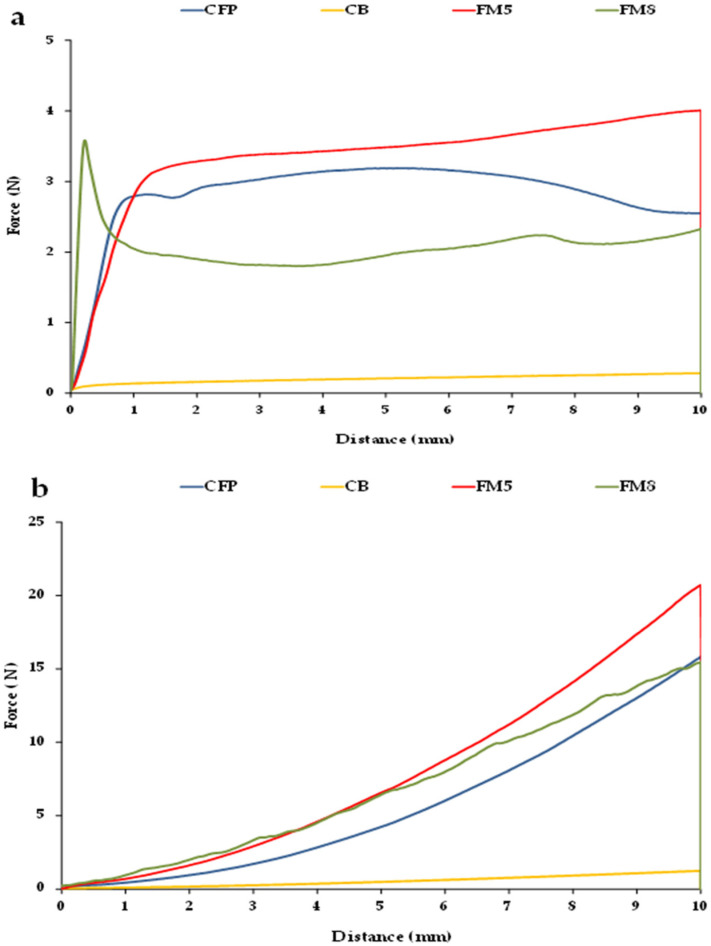
Force–distance curves at 20 °C of a commercial fatty preparation (CFP), a commercial butter (CB) and two formulated margarines (FM5 and FM8): (**a**) from the penetration test with a flat probe; (**b**) from the penetration test with a conical probe.

**Table 1 foods-11-04054-t001:** Composition, expressed as percentage (%), of the formulated margarines (FM) for puff pastry (PP).

Margarine	Aqueous Phase, AP (20%)			Oil Phase, OP (80%)			
	Water	Gelatine	MTG	OPO	Beeswax	Palsgaard 6111^®^	Verol N-90
FM1	18.5	-	-	35.4	4.5	0.90	-
FM2	18.5	-	-	40.0	-	-	0.83
FM3	17.2	1.5	-	40.0	-	-	0.83
FM4	18.0	-	0.75	40.0	-	-	0.83
FM5	18.5	-	-	35.0	5.0	-	0.83
FM6	18.5	-	-	37.0	3.0	-	0.83
FM7	18.5	-	-	40.0	-	0.80	-
FM8	18.5	-	-	40.0	-	0.90	-
FM9	18.6	-	-	40.0	-	0.70	-

All the AP margarines contained salt at 1.0%, citric acid at 0.17% and butter flavor at 0.33%. All the OP margarines contained fixed quantities of PS at 27.5%, butter at 7.5%, Verolec-Non GMO IP at 1.5%, Palsgaard 1311^®^ at 1.2%, and Verol P at 1.5%.

**Table 2 foods-11-04054-t002:** Critical (maximum) rheological properties at 20 °C on the limit of linear viscoelastic range (LVR) in the formulated margarines (FM) containing OPO in comparison with a commercial fatty preparation (CFP) and a commercial butter (CB) for puff pastry (PP).

Formulation	*σ*_max_ (kPa)	*γ*_max_ (%)	*G**_max_ (kPa)	tan *δ*_max_ (-)
CFP	1.00 ± 0.00 ^C^	0.0437 ± 0.0019 ^C,D^	2297 ± 100 ^C,D^	0.164 ± 0.0028 ^B^
CB	0.100 ± 0.00 ^H^	0.0663 ± 0.0072 ^A^	152 ± 16 ^G^	0.184 ± 0.00013 ^A^
FM1	0.216 ± 0.00 ^F^	0.0234 ± 0.0020 ^F,G^	1044 ± 79 ^F^	0.123 ± 0.015 ^C^
FM2	0.0502 ± 0.00 ^I^	0.0166 ± 0.00086 ^G^	303 ± 16 ^G^	0.117 ± 0.0015 ^C,D^
FM3	0.796 ± 0.00 ^D^	0.0393 ± 0.0085 ^C–E^	2125 ± 160 ^D^	0.0843 ± 0.0011 ^F^
FM4	0.159 ± 0.00 ^G^	0.0393 ± 0.0085 ^D–F^	504 ± 9.6 ^G^	0.132 ± 0.0014 ^C^
FM5	1.42 ± 0.00 ^A^	0.0508 ± 0.0095 ^B,C^	2892 ± 143 ^B^	0.102 ± 0.0019 ^D,E^
FM6	1.13 ± 0.00 ^B^	0.0418 ± 0.0026 ^C,D^	2700 ± 168 ^B,C^	0.102 ± 0.0020 ^D,E^
FM7	1.42 ± 0.00 ^A^	0.0602 ± 0.00013 ^A,B^	2350 ± 5.0 ^C,D^	0.0891 ± 0.0015 ^E,F^
FM8	1.13 ± 0.00 ^B^	0.0328 ± 0.0064 ^D–F^	3490 ± 680 ^A^	0.0750 ± 0.00070 ^F^
FM9	0.399 ± 0.00 ^E^	0.0660 ± 0.0084 ^A^	1621 ± 98 ^E^	0.0859 ± 0.0013 ^E,F^

Mean values (*n* = 3) ± standard deviation. *σ*_max_, critical value of shear stress; *γ*_max_, critical value of shear strain; *G**_max_, critical value of complex modulus (*G** = [(*G*′)^2^ + (*G*″)^2^]^0.5^); tan *δ*_max_, critical value of loss factor (*G*″/*G*′). ^A–I^ Different letters in the same column indicate significant differences (*p* < 0.05).

**Table 3 foods-11-04054-t003:** Mechanical spectra data at 1 Hz and at 20 °C, and weak-gel model parameters in margarines containing OPO in comparison with a commercial fatty preparation (CFP) and a commercial butter (CB) for puff pastry (PP).

Formulation	*G*′ (kPa)	*G*″ (kPa)	tan *δ*	*A* (kPa s^1/*z*^)	*z* (-)	*R* ^2^
CFP	2479 ± 26 ^E^	317 ± 2.8 ^A^	0.129 ± 0.00010 ^C^	2479 ± 26 ^E^	12.8 ± 0.24 ^F,G^	1.00
CB	156 ± 3.4 ^I^	23.6 ± 0.24 ^J^	0.151 ± 0.0021 ^B^	157 ± 3.7 ^J^	11.4 ± 0.58 ^G^	1.00
FM1	1224 ± 26 ^G^	136 ± 1.7 ^G^	0.111 ± 0.0015 ^D^	1234 ± 26 ^G^	16.4 ± 0.054 ^D,E^	1.00
FM2	237 ± 3.6 ^I^	38.6 ± 0.32 ^I^	0.163 ± 0.0013 ^A^	239 ± 3.3 ^I^	14.9 ± 1.0 ^E,F^	1.00
FM3	2805 ± 4.2 ^C^	174 ± 1.8 ^E^	0.0619 ± 0.00067 ^G^	2819 ± 1.6 ^C^	28.6 ± 1.4 ^B^	1.00
FM4	522 ± 8.3 ^H^	57.9 ± 0.31 ^H^	0.111 ± 0.0012 ^D^	524 ± 7.9 ^H^	17.4 ± 0.68 ^D^	1.00
FM5	2651 ± 18 ^D^	234 ± 3.9 ^C^	0.0882 ± 0.0011 ^E^	2657 ± 16 ^D^	18.3 ± 1.1 ^D^	1.00
FM6	3008 ± 66 ^B^	253 ± 0.66 ^B^	0.0843 ± 0.0016 ^E^	3014 ± 63 ^B^	21.5 ± 0.32 ^C^	1.00
FM7	2699 ± 15 ^D^	166 ± 4.1 ^F^	0.0613 ± 0.0019 ^G^	2707 ± 14 ^D^	27.0 ± 0.76 ^B^	1.00
FM8	3319 ± 53 ^A^	185 ± 5.1 ^D^	0.0558 ± 0.0014 ^H^	3337 ± 40 ^A^	31.5 ± 1.2 ^A^	0.992
FM9	2028 ± 32 ^F^	134 ± 1.0 ^G^	0.0659 ± 0.00054 ^F^	2038 ± 32 ^F^	27.8 ± 0.52 _B_	0.998

Mean values (*n* = 3) ± standard deviation. *G*’, elastic modulus; *G*″, viscous modulus; tan *δ*, loss factor (*G*″/*G*′); *G**, complex modulus; ^A–I^ Different letters in the same column indicate significant differences (*p* < 0.05).

**Table 4 foods-11-04054-t004:** Crossover temperature of viscoelastic moduli in margarines containing OPO in comparison with a commercial fatty preparation (CFP) and a commercial butter (CB) for puff pastry (PP).

Formulation	Crossover Temperature (°C)
CFP	41.0 ± 0.16 ^A^
CB	30.0 ± 0.27 ^G^
FM1	37.5 ± 1.9 ^D,E^
FM2	33.2 ± 0.95 ^F^
FM3	38.4 ± 0.72 ^C,D^
FM4	35.6 ± 1.2 ^E^
FM5	38.7 ± 0.92 ^B–D^
FM6	40.6 ± 0.56 ^A,B^
FM7	39.6 ± 0.20 ^A–C^
FM8	39.1 ± 0.035 ^B–D^
FM9	37.2 ± 0.89 ^D,E^

Mean values (*n* = 3) ± standard deviation. ^A–G^ Different letters indicate significant differences (*p* < 0.05).

**Table 5 foods-11-04054-t005:** Textural parameters derived from flat and conical penetration tests at 20 °C in margarines containing OPO in comparison with a commercial fatty preparation (CFP) and a commercial butter (CB) for puff pastry (PP).

	Flat Penetration			Conical Penetration	
Formulation	Force at 10 mm (N)	Maximum Force (N)	Work (mJ)	Force at 10 mm (N)	Work (mJ)
CFP	2.85 ± 0.055 ^B^	3.21 ± 0.013 ^C^	29.2 ± 0.36 ^C^	15.6 ± 0.41 ^B,C^	55.8 ± 2.4 ^C^
CB	0.300 ± 0.014 ^I^	0.300 ± 0.014 ^H^	2.18 ± 0.11 ^K^	1.25 ± 0.00058 ^I^	5.38 ± 0.020 ^H^
FM1	1.01 ± 0.016 ^G^	1.01 ± 0.016 ^F^	8.43 ± 0.084 ^H^	5.76 ± 0.12 ^F,G^	21.3 ± 0.43 ^F^
FM2	0.612 ± 0.010 ^H^	0.614 ± 0.011 ^G^	5.40 ± 0.061 ^J^	3.73 ± 0.10 ^H^	14.8 ± 0.068 ^G^
FM3	1.74 ± 0.034 ^E^	2.13 ± 0.056 ^D^	16.3 ± 0.26 ^E^	12.4 ± 0.13 ^D^	53.2 ± 0.72 ^C^
FM4	0.844 ± 0.067 ^G^	0.849 ± 0.060 ^F,G^	6.74 ± 0.34 ^I^	4.64 ± 0.057 ^G,H^	17.6 ± 0.53 ^F,G^
FM5	4.01 ± 0.10 ^A^	4.01 ± 0.11 ^A^	33.4 ± 0.49 ^A^	20.7 ± 0.26 ^A^	78.0 ± 1.5 ^A^
FM6	4.03 ± 0.14 ^A^	4.22 ± 0.14 ^A^	31.5 ± 0.49 ^B^	16.2 ± 0.29 ^B^	68.8 ± 0.41 ^B^
FM7	2.12 ± 0.061 ^D^	2.12 ± 0.061 ^D^	14.4 ± 0.21 ^F^	9.72 ± 0.64 ^E^	37.0 ± 2.1 ^D^
FM8	2.38 ± 0.043 ^C^	3.70 ± 0.13 ^B^	20.9 ± 0.32 ^D^	14.8 ± 0.55 ^C^	63.4 ± 4.2 ^B^
FM9	1.46 ± 0.039 ^F^	1.46 ± 0.039 ^E^	11.5 ± 0.51 ^G^	6.78 ± 0.22 ^F^	27.6 ± 0.91 ^E^

Mean values (*n* = 3) ± standard deviation. ^A–J^ Different letters in the same column indicate significant differences (*p* < 0.05).

**Table 6 foods-11-04054-t006:** Crystallization melting peak temperatures determined by DSC in palm stearin (PS), a commercial fatty preparation (CFP), a commercial butter (CB), and three margarines containing OPO.

Formulation	*T*_cp1_ (°C)	*T*_cp2_ (°C)	*T*_mp1_ (°C)	*T*_mp2_ (°C)	*T*_mp3_ (°C)
PS *	2.77 ± 0.38	34.4 ± 0.12	8.29 ± 0.19	44.9 ± 0.11	54.9 ± 0.26
CFP	1.67 ± 0.31	24.9 ± 1.3 ^B^	5.95 ± 0.20	42.9 ± 0.33 ^A^	-
CB *	9.56 ± 0.96	15.6 ± 0.25	−3.34 ± 1.2	14.5 ± 0.070	33.9 ± 0.40
FM5	-	26.1 ± 0.10 ^A,B^	−7.53 ± 0.64	41.8 ± 0.15 ^B^	-
FM7	-	26.9 ± 0.13 ^A^	−6.50 ± 1.9	42.1 ± 0.11 ^B^	-
FM9	-	25.7 ± 0.34 ^A,B^	−9.30 ± 0.66	41.7 ± 0.13 ^B^	-

Mean values (*n* = 3) ± standard deviation. DSC, differential scanning calorimetry; *T*_cp1_, *T*_cp2_, peak temperatures detected from crystallization thermograms; *T*_mp1_, *T*_mp2_, *T*_mp3_, peak temperatures detected from melting thermograms. ^A,B^ For each thermogram temperature, different letters in the same column indicate significant differences (*p* < 0.05). * Samples are not included in any statistical analysis.

**Table 7 foods-11-04054-t007:** Fatty acid (FA) profile corresponding to palm stearin (PS), a commercial fatty preparation (CFP), a commercial butter (CB) and two margarines (FM5 and FM8) containing OPO for puff pastry (PP).

Fatty Acids (mg FA/g Sample)	PS	CFP	CB	FM5	FM8
Butyric C4:0	ND	ND	10.7 ± 0.25	0.810 ± 0.017	0.816 ± 0.033
Caproic C6:0	ND	ND	11.4 ± 0.23	0.842 ± 0.036	0.828 ± 0.050
Caprylic C8:0	0.344 ± 0.0045	ND	9.0 ± 0.12	0.830 ± 0.012	0.856 ± 0.040
Capric C10:0	0.317 ± 0.0055	ND	22.4 ± 0.26	1.86 ± 0.016	1.88 ± 0.053
Lauric C12:0	3.60 ± 0.012	0.359 ± 0.011	30.2 ± 0.32	3.58 ± 0.052	3.70 ± 0.088
Myristic C14:0	10.5 ± 0.028	3.22 ± 0.066	94.5 ± 1.1	11.3 ± 0.074	11.4 ± 0.042
Pentadecylic C15:0	0.734 ± 0.013	0.238 ± 0.017	10.1 ± 0.13	1.14 ± 0.011	1.15 ± 0.0091
Palmitic C16:0	614 ± 1.0	220 ± 4.2	284 ± 3.8	287 ± 4.3	294 ± 6.1
Margaric C17:0	1.24 ± 0.0023	0.652 ± 0.035	5.48 ± 0.075	1.14 ± 0.019	1.15 ± 0.024
Stearic C18:0	67.7 ± 0.12	34.0 ± 0.55	84.3 ± 1.2	55.8 ± 0.83	56.6 ± 1.4
Arachidic C20:0	3.75 ± 0.026	8.43 ± 0.13	1.08 ± 0.066	4.38 ± 0.029	4.49 ± 0.024
Behenic C22:0	0.755 ± 0.034	4.96 ± 0.024	0.46 ± 0.0053	5.41 ± 0.17	6.19 ± 0.26
Lignoceric C24:0	0.739 ± 0.027	5.81 ± 0.16	ND	0.826 ± 0.031	0.838 ± 0.058
∑SFA	704 ± 1.1	278 ± 5.9	563 ± 7.5	375 ± 5.2	384 ± 7.5
Palmitoleic C16:1n7	0.579 ± 0.0049	4.40 ± 0.094	13.5 ± 0.16	5.48 ± 0.15	5.34 ± 0.15
Vaccenic C18:1n7	3.06 ± 0.099	11.5 ± 0.26	3.87 ± 0.072	11.9 ± 0.40	11.5 ± 0.27
Oleic C18:1n9	184 ± 0.43	392 ± 8.6	163 ± 2.0	411 ± 12	403 ± 10
∑MUFA	188 ± 0.49	408 ± 8.9	180 ± 2.2	428 ± 13	419 ± 11
Linoleic C18:2n6c	35.8 ± 0.14	76.7 ± 1.7	13.1 ± 0.56	78.0 ± 2.1	76.6 ± 1.82
Linolenic C18:3n3	0.482 ± 0.0047	3.81 ± 0.062	3.69 ± 0.095	4.19 ± 0.088	4.10 ± 0.093
∑PUFA	36.3 ± 0.14	80.5 ± 1.6	16.8 ± 0.62	82.2 ± 2.1	80.8 ± 1.9
∑MUFA + ∑PUFA	224	488	197	510	500

Mean values (*n* = 3) ± standard deviation. ND, not detected. FA, fatty acids; SFA, saturated fatty acids; MUFA, monounsaturated fatty acids; PUFA, polyunsaturated fatty acids.

## Data Availability

Data is contained within the article or Appendix A.

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
