# Peer review of "Development and Physico-Chemical Characterization of Healthy Puff Pastry Margarines Made from Olive-Pomace Oil"

_foods, 2022, doi:10.3390/foods11244054_

Round 1
Reviewer 1 Report
The manuscript has investigated the effects of different oleogelators on some physicochemical properties of olive oil-based puff pastry margarine. The topic is interesting and of great importance; however, the manuscript is complicated and difficult to understand.
Comments:
Please add a paragraph about puff pastry, the ingredients, preparation, the reason for using high contents of butter or margarine, and the importance of using a healthy fat in this product.
Figures 1 and 2: Please show different parameters in different graphs; the graphs are very confusing.
Author Response
The manuscript has investigated the effects of different oleogelators on some physicochemical properties of olive oil-based puff pastry margarine. The topic is interesting and of great importance; however, the manuscript is complicated and difficult to understand.
Comments:
Please add a paragraph about puff pastry, the ingredients, preparation, the reason for using high contents of butter or margarine, and the importance of using a healthy fat in this product.
About the Introduction of the manuscript, the reviewer 2 stated “The introduction describes the matter of the experiment and states the problem being investigated. Authors correctly interpreted and described the significance of the results for the research. They skillfully referred to the results of other researchers. Literature references are the most current”.
However, in order to meet with the reviewer 1, in the Introduction we have added the the following paragraphs about puff pastry:
“PP is an aerated and laminated product with a high fat content obtained by repeated lamination of wheat dough and fat [1]. The fat used should be solid and firm enough to separate the layers of dough during the lamination process.”
Respect to other aspects about the PP that the reviewer suggested to mention in the introduction, we think they are already covered in the original introduction. As in this work the PP are not made, we considered that the description of the PP is not be necessary in this manuscript. However, in a second work more focused on the production of PP that is being carried out in these days, the elaboration of these puff pastries will be described in detail.
Figures 1 and 2: Please show different parameters in different graphs; the graphs are very confusing.
In order to clarify and improve the quality of the Figures 1 and 2, several formulated margarines have been deleted from the same. In addition, the results from all the formulated margarines are included in the different Tables.
Reviewer 2 Report
Manuscript ID: Foods-2070708
In the article entitled: “Development and physico-chemical characterization of healthy puff pastry margarines made from olive-pomace oil”.
This work studies the potential of olive-pomace oil as main ingredient of margarines together with different organogelator agents to imitate the technological properties of a commercial fatty preparation. Generally the work is interesting. The obtained test results can be used in the food industry. The authors used advanced measuring techniques suitable for the adopted purpose of research.
Title
The title and the aim of the study are clearly constructed.
Abstract
The abstract includes the aim of the study, methods used in the experiment and contain the principal results and conclusions.
Introduction
The introduction describes the matter of the experiment and states the problem being investigated. Authors correctly interpreted and described the significance of the results for the research. They skillfully referred to the results of other researchers. Literature references are the most current.
Materials and Methods
The data is well collected. The sampling is appropriate and adequately described.
Results and Discussion
Although the authors correctly interpreted and described the significance of the results for the research, I feel a bit unsatisfied. This is due to the fact that the discussion is basically limited to the presentation of results. On the other hand, their interpretation is poor. In general, more discussions and comparisons with the literature can be added to the article.
Conclusion
The authors correctly indicate, how the results are related to the studies.
References
Literature references are appropriate and relate to the position from the last few years. Language
The article is correctly written. English language and style are minor spell check required.
Due to the comments on Results and Discussion, I believe that the paper should be improved.
Below are the detailed comments:
This evolution would be associated with reorganization, recovery, and even with a pos-263 sible structure crystallization of diverse crystalline entities, that is, of mixtures of lipids or 264 TG, which are present in the samples according to their different composition, and that 265 would have suffered a destabilization process. 266
263-266: Associated with the reorganization, … What fatty acids? All?
286-287: constituted by a more compact and stronger fat-crystal network.
What does it mean? A denser crystalline network, why?
Moreover, these four FM showed the highest 303 values of fat-crystal rigidity (G*max), being FM7 the margarine that had the most similar 304 value to that of CFP. In contrast, FM9 had the lowest σmax, which could be associated with 305 its lower TG content (Table 1).
303-306: four FM showed the highest values of fat-crystal rigidity (G*max),…
What caused this effect?
In contrast, FM9 had the lowest σmax, which could be associated with its lower TG content…
TG is an enzyme that catalyses protein crosslinking, so what is its interaction with fatty acids?
Finally, CFP and CB presented a tan δmax significantly higher than that of the FM, 332 thus evidencing the presence of an internal network structure with a lower viscoelasticity 333 or a lower degree of solidity.
332-334: What does it mean the presence of an internal network structure? Is there any external network?
This result would 406 reflect that butter presents a crystalline network that, although it seems to have a lower 407 degree of connectivity than that of the rest of the margarines (lower A and z values), 408
406-408: lower degree of connectivity than that of the rest of the margarines (lower A and z values), …
More clearly what it means to lower degree of connectivity.
Author Response
Comments and Suggestions for Authors
Manuscript ID: Foods-2070708
In the article entitled: “Development and physico-chemical characterization of healthy puff pastry margarines made from olive-pomace oil”.
This work studies the potential of olive-pomace oil as main ingredient of margarines together with different organogelator agents to imitate the technological properties of a commercial fatty preparation. Generally the work is interesting. The obtained test results can be used in the food industry. The authors used advanced measuring techniques suitable for the adopted purpose of research.
Thank you very much for your kind general comments related to our manuscript.
Title
The title and the aim of the study are clearly constructed.
Abstract
The abstract includes the aim of the study, methods used in the experiment and contain the principal results and conclusions.
Introduction
The introduction describes the matter of the experiment and states the problem being investigated. Authors correctly interpreted and described the significance of the results for the research. They skillfully referred to the results of other researchers. Literature references are the most current.
Materials and Methods
The data is well collected. The sampling is appropriate and adequately described.
Results and Discussion
Although the authors correctly interpreted and described the significance of the results for the research, I feel a bit unsatisfied. This is due to the fact that the discussion is basically limited to the presentation of results. On the other hand, their interpretation is poor. In general, more discussions and comparisons with the literature can be added to the article.
Conclusion
The authors correctly indicate, how the results are related to the studies.
References
Literature references are appropriate and relate to the position from the last few years. Language
The article is correctly written. English language and style are minor spell check required.
Due to the comments on Results and Discussion, I believe that the paper should be improved.
Results and Discussion have been improved in accordance with the suggested and detailed comments from reviewer 2.
Below are the detailed comments:
This evolution would be associated with reorganization, recovery, and even with a pos-263 sible structure crystallization of diverse crystalline entities, that is, of mixtures of lipids or 264 TG, which are present in the samples according to their different composition, and that 265 would have suffered a destabilization process. 266
263-266: Associated with the reorganization, … What fatty acids? All?
We think that the two principles fatty acids present in the formulated margarines, palm stearin and in the commercial fatty preparation (CFP) and commercial butter (CB) were mainly affected. It can be shown in Table 7 of the manuscript, both major dominant fatty acids are palmitic acid (C16:0) and oleic acid (C18:1n9), and they significantly determine the ratio of saturated/unsaturated fatty acids of these samples. A short mention has been introduced in Results and Discussion (lines 263-266).
286-287: constituted by a more compact and stronger fat-crystal network.
What does it mean? A denser crystalline network, why?
We agree with the reviewer 1. Effectively, “a more compact and stronger fat-crystal netwok” means “a denser crystalline network”. We have changed “more compact and stronger” to “denser” as suggested for reasons of clarity. The main reason why FM5 and FM7 had denser crystalline networks is because of their highest critical shear stress (σmax) which is related to their composition as it is explained in detail in the text (lines 287, 288).
Moreover, these four FM showed the highest 303 values of fat-crystal rigidity (G*max), being FM7 the margarine that had the most similar 304 value to that of CFP. In contrast, FM9 had the lowest σmax, which could be associated with 305 its lower TG content (Table 1).
303-306: four FM showed the highest values of fat-crystal rigidity (G*max),…
What caused this effect?
In the case of FM5 and FM6, their higher rigidity could be associated with the presence of an oleogel (prepared with OPO and beeswax at 5% and 3%, respectively) and MG from Verol N-90, as it was mentioned in lines 309-311. On the other hand, both FM7 and FM8 contain fully hydrogenated triglicerides (TG) from Palsgaard 6111® at 0.80 and 0.90%, respectively.
In contrast, FM9 had the lowest σmax, which could be associated with its lower TG content…
TG is an enzyme that catalyses protein crosslinking, so what is its interaction with fatty acids?
In this manuscript, the abbreviator “TG” means triglicerides, not the enzyme. In fact, the only sample, which contains transglutaminase from microbian origin (MTG) was FM4.
FM9 had the lowest σmax and G*max due to its lower fully hydrogenated triglicerides (TG) content from Palsgaard 6111® (0.70%). So, it seems that this lower percentage is not enough to obtain a dense crystalline network.
Finally, CFP and CB presented a tan δmax significantly higher than that of the FM, 332 thus evidencing the presence of an internal network structure with a lower viscoelasticity 333 or a lower degree of solidity.
332-334: What does it mean the presence of an internal network structure? Is there any external network?
We agree with the reviewer 1. It was used the term “internal network structure” to highlight the network structure but to avoid confusion the term “internal” has been eliminated.
This result would 406 reflect that butter presents a crystalline network that, although it seems to have a lower 407 degree of connectivity than that of the rest of the margarines (lower A and z values), 408
406-408: lower degree of connectivity than that of the rest of the margarines (lower A and z values), …
More clearly what it means to lower degree of connectivity.
As it was explained in the manuscript, from a rheological point of view, the connectivity of a network could be defined by the parameters “A” and “z” as Lupi et al. (2011) and Gabriele et al., 2001). The parameter “z” is the network extension, related to the number of interacting rheological units within the 3-D network, and “A” is the strength of the interactions. When A increases, the interaction forces within the network increase, whereas a high z value indicates a large number of interacting units cooperating and increasing the network connectivity.
Therefore, as the butter showed the lowest both A and z values, it showed a lower degree of connectivity.
Round 2
Reviewer 1 Report
The manuscript is acceptable.
Author Response
It was made previously.
Reviewer 2 Report
The work takes into account the reviewer's comments.
Author Response
It was made previously.